# PoE: Process of Elimination for Multiple Choice Reasoning

**Chenkai Ma**[1] and **Xinya Du**[2]

[1]School of Computer Science and Engineering,
University of Electronic Science and Technology of China
[2]Deparment of Computer Science, University of Texas at Dallas
kasmas316@gmail.com    xinya.du@utdallas.edu

## Abstract

Language models (LMs) are capable of conducting in-context learning for multiple choice reasoning tasks, but the options in these tasks are treated equally. As humans often first eliminate wrong options before picking the final correct answer, we argue a similar two-step strategy can make LMs better at these tasks. To this end, we present the Process of Elimination (PoE), a two-step scoring method. In the first step, PoE scores each option, and eliminates seemingly wrong options. In the second step, PoE masks these wrong options, and makes the final prediction from the remaining options. Zero-shot experiments on 8 reasoning tasks illustrate the effectiveness of PoE, and a following analysis finds our method to be especially performant on logical reasoning tasks. We further analyze the effect of masks, and show that PoE applies to few-shot settings and large language models (LLMs) like ChatGPT.[1]

## 1 Introduction

> How often have I said to you that when you have eliminated the impossible whatever remains, however improbable, must be the truth? (Doyle, 1890)

In natural language processing, many reasoning tasks are multiple choice-based, in which a model chooses the best option from several options, given a question. Current LMs exhibit remarkable performance on diverse reasoning tasks, like commonsense reasoning (Wang et al., 2023a; Holtzman et al., 2021), logical reasoning (Ye et al., 2023), and arithmetic reasoning (Shum et al., 2023).

There are two types of in-context learning methods for multiple choice reasoning: scoring and prompting. Given the question, scoring methods score each option, and select the highest-scored

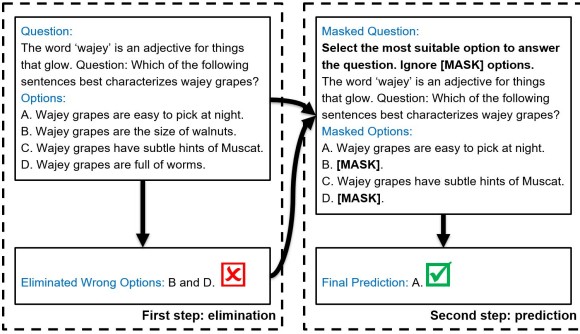

Figure 1: An illustration of the Process of Elimination (PoE) for a multiple choice question. In the first step, PoE eliminates some wrong options. In the second step, it enforces the masks, and makes the final prediction.

option. One common score is language modeling likelihood ($P(y_i|x)$, Brown et al., 2020), and there are also many alternatives (Zhao et al., 2021; Holtzman et al., 2021; Fei et al., 2023; Min et al., 2022; Malkin et al., 2022; Robinson and Wingate, 2023; Lyu et al., 2023). Tailored for powerful LLMs (Chowdhery et al., 2022; Chen et al., 2021), prompting methods (Wei et al., 2022; Wang et al., 2023b; Kojima et al., 2022; Besta et al., 2023; Yao et al., 2023) append all options to the question, and prompt the model to generate a sequence, then extract the option from the sequence.

One limitation of these two types of approaches is that they both treat each option equally, i.e., they either consider each option independently, or consider all options jointly. On the contrary, when humans solve a multiple choice reasoning task, they often eliminate some wrong options, then choose from the remaining ones. This so-called process of elimination is widely used in college exams and can be stronger than usual Gold style learning (Freivalds et al., 2002). Therefore, we assume that language models can similarly benefit from this elimination process, i.e., eliminating wrong options and choosing the best option are two types of reasoning that can be disentangled into two steps.

---

[1]Code is available at https://github.com/KasMasVan/PoE.

To this end, we present the Process of Elimination (PoE), a two-step scoring method for multiple choice reasoning, as shown in Figure 1. In the first step, PoE scores each option, then eliminates some wrong options based on their scores. In the second step, it masks these wrong options, then chooses the best one from the remaining options. We conduct experiments on 8 reasoning tasks that cover diverse domains. PoE achieves the best zero-shot performance on most tasks. A following analysis shows that it favors logical reasoning tasks. We also measure the effect of masks, and find our method applicable to few-shot settings and compatible with LLMs like ChatGPT (Ouyang et al., 2022).

Our **contributions** are twofold: (1) We present PoE, a two-step scoring method for multiple choice reasoning; (2) Through comprehensive experiments and analysis, we demonstrate the effectiveness and generalizability of PoE.

## 2 Method

**Problem Setting.** A multiple choice reasoning task instance includes a question $x$, several options $Y = \{y_1, ..., y_n\}$, and the correct option $y$. There are two kinds of in-context learning approaches to this problem: scoring and prompting.

Scoring uses an LM to compute a score for each option $y_i$, and chooses the highest-scored option:

$$s_i = \text{score}(x, y_i), \tag{1}$$

$$\hat{y} = \arg\max_i s_i. \tag{2}$$

A commonly-used score is language modeling likelihood ($P(y_i|x)$, Brown et al., 2020), More recent scores include average log probability (Brown et al., 2020), calibrated log probability (Holtzman et al., 2021), and channel (Min et al., 2022). As shown in Equation 1, most scoring methods consider each option in isolation, except multiple choice prompting (Robinson and Wingate, 2023).

In contrast, prompting methods provide all the options in the input (Wei et al., 2022; Wang et al., 2023b; Kojima et al., 2022). The model then generates raw output from the input. Finally, these methods extract the option from the raw output:

$$\text{raw output} = \text{generate}(x, Y), \tag{3}$$

$$\hat{y} = \text{extract}(\text{raw output}). \tag{4}$$

**PoE.** Our method is a two-step scoring method that considers all options but treats them differently.

We also implement a prompting-based variation of PoE for LLM (Section 5).

The first step of PoE is **elimination**, in which it eliminates some wrong options. PoE starts by scoring each option. Then, unlike common scoring methods that choose the highest-scored option, it uses the scores to eliminate some options with low scores. In particular, PoE computes the average score of all options, and eliminates options whose score is below average, i.e., $Y_{\text{wrong}}$:

$$Y_{\text{wrong}} = \{y_i | s_i < \text{avg}(s_1, ..., s_i)\}. \tag{5}$$

The intuition behind this elimination strategy ("Below Average") is that the scores of wrong options deviate from others, and are thus easy to identify, which we verify in Appendix B.2. We compare other elimination strategies in Section 5.2.

The second step of PoE is **prediction**, which chooses the best answer that is not in $Y_{\text{wrong}}$. Specifically, PoE computes binary masks $m_i$ for options:

$$m_i = \begin{cases} 0, & \text{if } y_i \in Y_{\text{wrong}} \\ 1, & \text{otherwise} \end{cases} \tag{6}$$

$$\text{mask} = [m_1, ..., m_n]. \tag{7}$$

For these options, PoE enforces the masks.[2] In particular (shown in Figure 1), it uses a template $T$ to first wrap the question with all options and their symbols like "A", then append an instruction to the question which asks the model to neglect masked options, and finally replace eliminated options with a special text sequence "[MASK]":

$$x_{\text{mask}} = T(x, Y, \text{mask}). \tag{8}$$

Then, PoE scores each option, and chooses the highest-scored option, during which the scores of eliminated options are set to negative infinity:

$$s_{\text{mask},i} = \begin{cases} \text{score}(x_{\text{mask}}, y_i), & \text{if } y_i \notin Y_{\text{wrong}} \\ -\text{inf}, & \text{otherwise} \end{cases} \tag{9}$$

$$\hat{y} = \arg\max_i s_{\text{mask},i}. \tag{10}$$

---

[2]We don't remove wrong options because that requires extra work like relabeling remaining options (e.g., "ADE" to "ABC"), but brings negligible difference to performance.

| Task | Method | | | | | |
|------|--------|-----|-------------|---------|------|------|
|      | LM | AVG | Calibration | Channel | MCP | PoE |
| ANLI | $38.6_{2.9}$ | $38.0_{3.4}$ | $37.2_{3.5}$ | $36.0_{2.7}$ | $\mathbf{57.8}_{3.0}$ | $\underline{55.0}_{2.4}$ |
| CQA | $64.4_{5.9}$ | $54.2_{1.3}$ | $69.6_{4.8}$ | $44.6_{5.6}$ | $\underline{87.2}_{2.6}$ | $\mathbf{89.2}_{2.2}$ |
| SIQA | $56.0_{4.5}$ | $62.6_{1.8}$ | $57.6_{6.9}$ | $39.8_{5.9}$ | $\underline{79.0}_{5.6}$ | $\mathbf{82.0}_{6.7}$ |
| LD | $\underline{48.8}_{2.7}$ | $45.8_{4.5}$ | $39.2_{4.0}$ | $20.8_{3.6}$ | $39.8_{3.7}$ | $\mathbf{53.6}_{7.0}$ |
| DQA | $45.8_{6.6}$ | $51.8_{2.0}$ | $48.0_{2.9}$ | $39.6_{6.6}$ | $\mathbf{67.8}_{1.9}$ | $\underline{67.4}_{2.5}$ |
| CC | $44.8_{0.8}$ | $51.8_{0.8}$ | $54.4_{0.5}$ | $44.0_{0.7}$ | $\underline{60.2}_{1.5}$ | $\mathbf{72.2}_{0.8}$ |
| SS | $39.0_{2.5}$ | $40.6_{1.8}$ | $43.4_{1.7}$ | $26.6_{1.7}$ | $\underline{74.0}_{0.7}$ | $\mathbf{76.6}_{0.9}$ |
| SIT | $24.8_{1.9}$ | $20.6_{5.3}$ | $21.0_{3.9}$ | $17.0_{4.6}$ | $\mathbf{25.4}_{2.9}$ | $\underline{25.2}_{4.0}$ |

Table 1: Accuracy scores (with standard deviation) on 8 tasks. The best scores are **boldfaced**, and the second-best scores are underlined. LM refers to language modeling, AVG refers to average language modeling, MCP refers to multiple choice prompting. Our method (PoE) achieves the best or comparable performance on all tasks.

## 3 Experiment Setup

**Data.** We consider 8 multiple choice reasoning tasks to cover diverse domains. We include three traditional reasoning tasks: ANLI (Nie et al., 2020), CommonsenseQA (CQA, Talmor et al., 2019), and Social IQa (SIQA, Sap et al., 2019). We also include five BIG-bench tasks (Srivastava et al., 2023), with the first two from BIG-Bench Hard (Suzgun et al., 2023): Logical Deduction (LD), Disambiguation_QA (DQA), Conceptual Combinations (CC), Strange Stories (SS), and Symbol Interpretation Task (SIT). For all tasks, we use their test sets if available, otherwise their development sets. To reduce cost, we sample 100 instances from each task. We present more task information and preprocessing in Appendix A.

**Model.** We use FLAN-T5-XL (3B) (Chung et al., 2022) in both steps of PoE, because it is an economical and performant instruction-tuned model. In Section 5, we also apply PoE to LLMs like ChatGPT (Ouyang et al., 2022).[3]

**Methods.** We consider 5 scoring baselines: language modeling (LM, baseline in Zhao et al., 2021), average language modeling (AVG, Brown et al., 2020), calibration (Holtzman et al., 2021), channel (Min et al., 2022), and multiple choice prompting (MCP, Robinson and Wingate, 2023). For PoE, we use MCP in both steps. We discuss other possible scoring methods and elimination strategies for PoE in Section 5.2, and present input and output samples for all methods in Appendix C.[4]

**Settings.** We evaluate PoE in the zero-shot setting. We use accuracy as the metric, and average results over 5 random seeds. We consider other settings in Section 5.3 and 5.4.

## 4 Results

In Table 1, we compare PoE with other baselines on 8 reasoning tasks, and present the accuracy and standard deviation. Our method achieves the best or second-best performance on all 8 tasks. We notice that multiple choice prompting (MCP) is a strong baseline that consistently outperforms other baselines. Nevertheless, PoE beats MCP on 5 tasks, and is comparable on the remaining three. Since PoE uses MCP in both steps, the result demonstrates the effectiveness of the former, especially elimination. In addition, PoE is exceptionally performant on CC, beating the second-best method by 12% absolute. This corroborates our hypothesis on two types of reasoning for multiple choice tasks, because while MCP sometimes fails to simultaneously eliminate wrong options and choose the right option, PoE disentangles the two jobs in two steps, and thus achieves better performance. We present a case study in Appendix B.2.

## 5 Analysis

### 5.1 When Is PoE Better than MCP?

In Table 1, we find MCP to be the best baseline, and sometimes even beats PoE. This finding leads us to wonder: when (on which tasks) can we expect PoE to dramatically outperform MCP?

---

[3] We do not use recent larger instruction-following models based on LLaMA (Touvron et al., 2023), because 1) they are not optimized for traditional NLP tasks, and 2) they evolve quickly, which makes it hard for a fair comparison.

[4] We do not include prompting baselines like Chain-of-Thought Prompting (Wei et al., 2022), because they are tailored for very large (e.g., 175B) models.

| Task | Method | | |
|------|--------|--------|-----------|
| | MCP | POE | POE - MCP |
| LA | 50.0 | 68.8 | +18.8 |
| IMT | 34.0 | 47.2 | +13.2 |
| CLD | 67.2 | 75.9 | +8.7 |
| RACO | 53.8 | 60.6 | +6.8 |
| CAI | 84.1 | 81.8 | -2.3 |
| EIE | 25.0 | 19.1 | -5.9 |
| RS | 55.1 | 49.0 | -6.1 |
| IOM | 56.2 | 50.0 | -6.2 |

Table 2: Comparison of MCP and POE accuracy scores on 8 new tasks. The top 4 tasks are logical reasoning tasks. POE largely outperforms MCP on 4 logical reasoning tasks, and underperforms MCP on other 4 tasks.

We start by calculating their performance gaps (POE - MCP) on all 8 tasks. We find the gaps on LD (13.8%) and CC (12%) are much larger than those of other tasks (less than 3%). Furthermore, we find both LD and CC are logical reasoning tasks, while other tasks rely more on commonsense or social reasoning.[5] We thus hypothesize that POE is better than MCP on tasks that mainly require logical reasoning, while not being consistently performant on social and commonsense reasoning. One explanation is that logical reasoning is a general skill, whereas social or commonsense reasoning relies on specific knowledge, some of which may be absent in certain language models.

To verify this hypothesis, we compare POE and MCP in the zero-shot setting on 8 new tasks from BIG-Bench, including 4 logical reasoning tasks: Logical Arguments (LA), Identify Math Theorems (IMT), Code Description (CLD), Reasoning about Colored Objects (RACO); and 4 control tasks that are based on commonsense or social reasoning: Counterfactual Conditionals (CAI), The Essential, the Excessive, and the Extraneous (EIE), Riddle-Sense (RS), Identify Odd Metaphor (IOM). We present task information in Appendix A.

As shown in Table 2, we find POE consistently and dramatically outperforms MCP on all 4 logical reasoning tasks (top 4 tasks). We find the largest performance gap on LA (18.8%), a task of GRE-style logical questions. Human test-takers commonly use elimination-based methods to solve such GRE questions, which further supports the motiva-

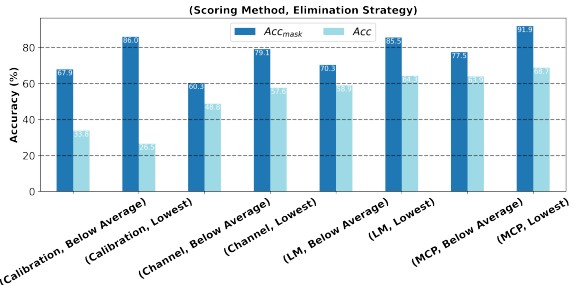

Figure 2: Effect of different scoring methods and elimination strategies on masking accuracy ($Acc_{mask}$) and final accuracy (Acc). The best configuration is (MCP, Lowest), with a large gap between $Acc_{mask}$ and Acc.

tion of POE. We also find it underperforms MCP on 4 control tasks (bottom 4 tasks), which means our method is not consistent on commonsense or social reasoning tasks. These findings support our hypothesis that compared to MCP, POE is most performant on logical reasoning tasks.

## 5.2 What Makes Good Masks?

As shown in Section 2, scoring methods and elimination strategies jointly determine which options to eliminate ($Y_{wrong}$), and create masks accordingly. In this section, we analyze the effect of using different configurations of these two factors.[6] We define masking accuracy ($Acc_{mask}$) as the ratio of instances that have their correct option kept after elimination. $Acc_{mask}$ represents the upper bound of POE, which we aim to maximize to reduce error propagation (Du and Cardie, 2020). We measure two quantities: $Acc_{mask}$ and final accuracy (Acc). We consider four scoring methods (Calibration, Channel, LM, MCP) and two elimination strategies ("Below Average" and "Lowest", the latter of which means eliminating the option with the lowest score). We average zero-shot results over all 8 tasks in Table 1.

As shown in Figure 2, the best scoring method is MCP, and the two elimination strategies are comparable to each other. We find that higher $Acc_{mask}$ leads to higher Acc. The best configuration of scoring method and elimination strategy (MCP, Lowest) leads to 91.9% $Acc_{mask}$.[7] This means a moderate-sized LM (FLAN-T5-XL) can eliminate wrong op-

---

[5]SIT is another logical reasoning task, but it also requires visual reasoning, which is not ideal for language models.

[6]We also tried mask tokens other than "[MASK]", but did not find one consistently better than others. So we don't tune this token to avoid unnecessary costs.

[7]We use (MCP, Below Average) in the main experiment, because this configuration makes POE more separable on tasks that it performs well, i.e., the gap between POE and the second-best baseline is larger.

| Task | Method | | |
|------|--------|------|-----------|
| | MCP | PoE | PoE - MCP |
| SIT | 2.0 | 19.4 | +17.4 |
| ANLI | 40.2 | 44.6 | +4.4 |
| SS | 79.4 | 82.8 | +3.4 |
| CQA | 75.0 | 77.0 | +2.0 |
| LD | 37.6 | 38.0 | +0.4 |
| SIQA | 72.6 | 71.2 | -1.4 |
| DQA | 48.8 | 42.6 | -6.2 |
| CC | 79.6 | 66.8 | -12.8 |
| RACO | 4.8 | 50.4 | +45.6 |
| IMT | 34.0 | 49.8 | +15.8 |
| LA | 50.0 | 55.0 | +5.0 |
| CLD | 92.4 | 91.4 | -1.0 |

Table 3: PoE's zero-shot performance on ChatGPT (`gpt-3.5-turbo-0613`). The bottom 4 tasks are logical reasoning tasks from Section 5.1. PoE consistently outperforms MCP, especially on logical reasoning tasks.

tions while keeping the correct ones most of the time. The corresponding 68.7% Acc means a large performance gap, which suggests a more powerful LM like FLAN-T5-XXL for prediction. We also find $Y_{\text{wrong}}$ and masks to make PoE more interpretable, because they provide intermediate reasoning outputs. Enforcing masks in prediction also makes our method faithful and factual.

### 5.3 Does PoE Work with LLMs?

To measure whether PoE is compatible with LLMs, we implement a prompting-based variation of PoE and apply it to ChatGPT (`gpt-3.5-turbo-0613`). We then measure its zero-shot performance on 8 original tasks in Table 1 and 4 logical reasoning tasks in Table 2. For PoE, we only use ChatGPT in the second step (prediction), as the base LM suffices for masking (Section 5.2). Concretely, we prompt ChatGPT to complete $x_{\text{mask}}$, and extract the last occurrence of any option symbol as the prediction. We compare PoE to MCP, which is similarly modified to work with ChatGPT.

We present the result in Table 3. The result from ChatGPT is consistent with those from FLAN-T5-XL (Table 1): PoE beats MCP on 5 out of the 8 original tasks, and performs well on 4 logical reasoning tasks. These findings suggest our method also works with LLMs. Nevertheless, we find ChatGPT underperforms FLAN-T5-XL on some tasks, which requires further experiments and is beyond

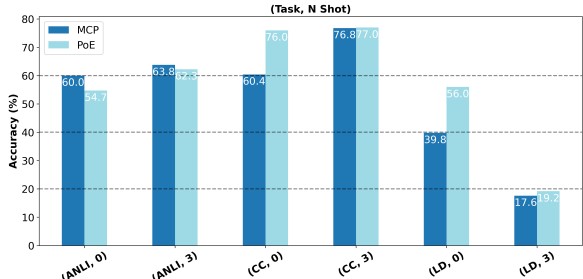

Figure 3: Three-shot and zero-shot accuracy scores on ANLI, CC, and LD. Although PoE shows mixed performance, it is applicable to few-shot settings.

the scope of this work.

### 5.4 Does PoE Work in Few-Shot Settings?

To measure PoE in the few-shot settings, we compare its zero-shot and three-short performance with MCP on ANLI, CC, and LD. We build three-shot demonstrations by randomly sampling from the training sets of ANLI and test sets of CC and LD.[8]

We present the result in Figure 3. For ANLI, PoE underperforms MCP in both settings, but their performance gap drops from 5.3% (zero-shot) to 1.5% (three-shot). For LD and CC, PoE outperforms MCP in both settings, but their performance gap similarly diminishes in the three-shot setting. Comparing PoE with MCP, We find three-shot PoE to be less performant than its zero-shot counterpart, and possible reasons include: 1) we did not optimize prompts or demonstrations for three-shot PoE; 2) the instructions we use in zero-shot PoE are already powerful, and three-shot demonstrations may introduce some noise. Still, our findings suggest PoE is applicable to few-shot settings, which we will continue to explore in future work.

### 6 Conclusion

We present PoE, a two-step scoring method for multiple choice reasoning. PoE eliminates some wrong options in the first step, and chooses from the remaining options in the second step. PoE performs well on reasoning (especially logical reasoning) tasks in the zero-shot setting, and also works with LLMs or in the few-shot setting. In the future, we will improve the generalizability of our method, and use fine-tuning to better enforce the masks.

---

[8]For CC and LD, we run experiments on the remaining data after sampling.

## Limitations

The first limitation of our work is the enforcement of the masks. Although we replace eliminated options with "[MASK]", and also replace their scores afterward, the model still considers these options in prediction. It would be better if the model does not consider these options at all. A second limitation is that we do not fully explore the components of PoE: We fix the prompts, which may not be optimal. A third limitation of our work is the scope of the LLM experiment: We only test in the zero-shot setting. It would be better if we also run few-shot LLM experiments, and incorporate complex prompting methods like Chain-of-Thought Prompting (Wei et al., 2022). A fourth limitation is that this work considers only one modality, i.e., language. It would be interesting to extend PoE to tasks involving other modalities, like vision (Seo et al., 2022).

## Ethics Statement

The tasks and models in this work are publicly available. They could contain bias, and should be used with discretion.

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

## A  Task Information and Pre-Processing

All tasks we use are expressed in English. ANLI is originally a classification task, and we convert it to multiple choice by mapping labels to options, i.e., "0" to "entailment", "1" to "neutral", and "2" to "contradiction". For BIG-bench tasks, we remove instances whose number of options deviate from others. For ANLI and CC, we aggregate instances from all subtasks; For LD, we use the subtask with

5 options; For SS, we use the multiple choice sub-task, and treat options with scores of 0.5 as wrong. We provide other details in Table 4. The first three tasks can be accessed on Hugging Face.[9] The last thirteen tasks can be accessed on BIG-bench.[10]

| Task | # Options | Train | Dev | Test |
|------|-----------|-------|-----|------|
| ANLI | 3 | 162865 | 3200 | 3200 |
| CQA | 5 | 9741 | 1221 | N/A |
| SIQA | 3 | 33410 | 1954 | N/A |
| CAI | 4 | N/A | N/A | 44 |
| CC | 4 | N/A | N/A | 103 |
| CLD | 4 | N/A | N/A | 60 |
| DQA | 3 | N/A | N/A | 255 |
| EIE | 5 | N/A | N/A | 68 |
| IMT | 4 | N/A | N/A | 53 |
| IOM | 4 | N/A | N/A | 47 |
| LA | 5 | N/A | N/A | 32 |
| LD | 5 | N/A | N/A | 500 |
| RACO | 18 | N/A | N/A | 2000 |
| RS | 5 | N/A | N/A | 49 |
| SIT | 5 | N/A | N/A | 990 |
| SS | 4 | N/A | N/A | 121 |

Table 4: Task Information. # Options means the number of options for the task.

# B   Additional Analysis

## B.1   Different Number of Options

Since our method eliminates wrong options in the first step, we assume the number of options affects performance. Consequently, we run experiments on three subtasks of logical deduction (LD (3), LD (5), LD (7)), which have 3, 5, and 7 options respectively. The other experiment settings are consistent with the main experiment (Section 3).

We present the result in Table 5. We find most methods (including PoE) perform worse as the number of options increases. This conforms to intuition, because more options require more reasoning steps, and the questions get harder. Nevertheless, PoE is the best on all subtasks. This means although PoE does not counterintuitively perform better on harder questions, it works well on questions of different difficulties, and thus applies to a variety of multiple choice reasoning tasks.

[9] https://huggingface.co/
[10] https://github.com/google/BIG-bench

| Method | Task | | |
|--------|------|------|------|
| | LD (3) | LD (5) | LD (7) |
| LM | $57.0_{3.9}$ | $48.8_{2.7}$ | $47.2_{2.7}$ |
| AVG | $55.8_{2.6}$ | $45.8_{4.5}$ | $47.6_{4.6}$ |
| Calibration | $45.0_{3.1}$ | $39.2_{4.0}$ | $38.8_{5.1}$ |
| Channel | $40.0_{3.8}$ | $20.8_{3.6}$ | $18.2_{2.2}$ |
| MCP | $53.4_{5.3}$ | $39.8_{3.7}$ | $45.2_{7.3}$ |
| PoE | $\mathbf{70.2}_{4.2}$ | $\mathbf{56.0}_{3.8}$ | $\mathbf{53.0}_{5.5}$ |

Table 5: Accuracy scores on three subtasks of logical deduction, which have different numbers of options. Best scores are **bold**, and second-best scores are underlined. PoE applies to tasks of varying difficulties.

## B.2   Case Study

In Figure 4, we present a case study to show why PoE works. We compare PoE with MCP on one instance from CC. The question invents the word "wajey" with a definition, and forms a surprising and uncommon conceptual combination between "wajey" and "grape". The correct option is A.

Question:
The word 'wajey' is an adjective for things that glow.
Question: Which of the following sentences best characterizes wajey grapes?
Options:
A. Wajey grapes are easy to pick at night.
B. Wajey grapes are the size of walnuts.
C. Wajey grapes have subtle hints of Muscat.
D. Wajey grapes are full of worms.

| Options | MCP Scores | PoE Scores |
|---------|-----------|-----------|
| A | 1.1426 | 0.6382 ✅ |
| B → [MASK] | 2.9434 | Inf |
| C | 0.8018 ❌ | 1.2979 |
| D → [MASK] | 4.6562 | Inf |

Figure 4: A case study comparing PoE and MCP. The scores are negative log-likelihood (lower means better). MCP eliminates options B and D, but gets distracted by option C. In contrast, PoE eliminates options B and D in the first step, and chooses the correct option A in the second step when B and D are masked.

We find MCP unable to solve this task, and chooses C. Taking a closer look at MCP scores (lower is better) for each option, we find that MCP assigns high scores to options B and D, two seemingly wrong options. This verifies our hypothesis that some wrong options' scores deviate from others, and can thus be eliminated. Nevertheless, MCP is distracted by option C, and we assume this

is caused by the correlation between "grape" and "Muscat", the latter of which is a kind of grape. This instance shows although MCP can eliminate some wrong options, it struggles to **simultaneously** choose the right option.

For PoE, since it uses MCP in the first step, it masks options B and D. We examine the final PoE scores for each option, and find that it correctly chooses option A. In addition, we find the PoE score for option A is lower than its MCP score, and conversely for option C. This means our method is not distracted by option C. This case study verifies our assumption that eliminating wrong options and choosing the best option are two types of reasoning that can be disentangled into two steps, and that such disentanglement is beneficial.

## C   Input and Output Samples for All Methods

We do not optimize prompts, because our goal is to evaluate PoE with a fixed prompt format. Therefore, we follow Holtzman et al. (2021) and Robinson and Wingate (2023) to construct task-agnostic prompts, and use them to wrap questions. We use one instance from SIQA. The question $x$ is "Kendall was searching for ring with their eyes closed. They hit something. Why did Kendall do this?". The options $Y$ are "kendall who has searching his ring" ($y_1$), "kendall who has wanted to close their eyes" ($y_2$), and "find the rings" ($y_3$).

We present input and output samples for all methods for this instance in Table 6. Calibration computes two scores ($\log P(y_i|x)$) and ($\log P(y_i|\text{text})$), so we present samples for each of them. PoE uses different prompts for each step, so we present them separately.

| Method | Score | Input | Output |
|---|---|---|---|
| LM | $\log P(y_i\|x)$ | $x$ the answer is: | $y_1$ |
| AVG | $1/\mathrm{len}(y_i) * \log P(y_i\|x)$ | $x$ the answer is: | $y_1$ |
| Calibration | $\log P(y_i\|x)$ | $x$ the answer is: | $y_1$ |
| | $\log P(y_i\|\text{text})$ | the answer is: | $y_1$ |
| Channel | $\log P(x\|y_i)$ | $y_1$ | $x$ the answer is: |
| MCP | $\log P(y_i\|x, Y)$ | Question: $x$
A. $y_1$
B. $y_2$
C. $y_3$
Answer: | A |
| PoE: Elimination | $\log P(y_i\|x, Y)$ | Question: $x$
A. $y_1$
B. $y_2$
C. $y_3$
Answer: | A |
| PoE: Prediction | $\log P(y_i\|x_{\text{mask}}, Y \setminus Y_{\text{wrong}})$ | Select the most suitable option to answer the question. Ignore [MASK] options.
Question: $x$
A. [MASK]
B. $y_2$
C. $y_3$
Answer: | B |

Table 6: Input and output samples for all methods, using one instance from SIQA.