# OpenReview forum: "POE: Process of Elimination for Multiple Choice Reasoning"
_EMNLP/2023/Conference — EMNLP 2023 Main_

### Official Review · Reviewer_dmiM · 2023-07-20

**Typos Grammar Style And Presentation Improvements:** 133
**Soundness:** 4

**Excitement:**

3: Ambivalent: It has merits (e.g., it reports state-of-the-art results, the idea is nice), but there are key weaknesses (e.g., it describes incremental work), and it can significantly benefit from another round of revision. However, I won't object to accepting it if my co-reviewers champion it.

**Paper Topic And Main Contributions:**

This paper contributes a strategy to boost multiple-choice QA accuracy for language models by first eliminating low-score answers then rescoring the top-scoring answers. They also provide an analysis of different design choices for this procedure.

The authors hypothesize that the MCQA requires two skills: eliminating choices and choosing the correct choice. Their results give evidence for the claim by showing PoE usually performs comparably with MCP for most tasks, but for some tasks beats MCP by a large margin.

**Questions For The Authors:**

1. Why do you mask the incorrect answers rather than simply removing them from the prompt?
2. Do you have any idea why the top-choice ordering changes in some cases after elimination?

**Reasons To Accept:**

This paper is clearly written and shows promising results for their method. Their contribution seems fairly scoped for a short paper. I appreciate their thoroughness in considering the space of design decisions for their method.

**Reasons To Reject:**

What would make this paper interesting to me would be to have some idea of *when* PoE can be expected to dramatically outperform MCP (i.e., what makes LD and CC tasks unique). Why does the scoring order change after masking? Without answering, analyzing, or addressing this question the contribution seems uninformative. For instance, in Holtzman et al. (2021) the authors provide a hypothesis for a mechanism for exactly how their method works (surface form competition).

A more easily resolved criticism I have is that, since MCP is the closest method to the paper's method, the authors should show a clear advantage to using PoE over MCP in some cases. Therefore, in Table 2 I would be more interested in seeing the results for LD or CC to see if the large gains from using PoE for these tasks persists in the few-shot setting. This should be easily addressable with an added experiment.

**Reproducibility:**

5: Could easily reproduce the results.

**Reviewer Confidence:**

4: Quite sure. I tried to check the important points carefully. It's unlikely, though conceivable, that I missed something that should affect my ratings.

---

> ### Author Rebuttal · Authors · 2023-08-28
>
> We are truly grateful for the constructive feedback from this review. We are excited to find that the reviewer appreciates our method, experiments and analysis, and the clarity and reproducibility (5) of the paper. We are delighted to find our paper “fairly scoped for a short paper”.
>
> # Reasons to Reject:
>
> > 1. *When* PoE [our method] can be expected to dramatically outperform MCP [best baseline] (i.e., what makes LD and CC tasks unique [our method is really good on these two tasks])
>
> **When:** PoE dramatically outperforms MCP on tasks that mainly require logical reasoning, but not on tasks about social or commonsense reasoning.
>
> **Observation**: Both tasks (LD and CC) are based on logical reasoning. Among the 8 tasks we used in our experiment, there is one more logical reasoning task: SIT, on which MCP is comparable to PoE. Nevertheless, SIT relies on visual reasoning, which may weaken PoE.
>
> **Hypothesis**: PoE is better than MCP on tasks that mainly require logical reasoning. In addition, PoE is not consistently good at social or commonsense reasoning. One explanation is that logical reasoning is a general skill, whereas social or commonsense reasoning relies on specific knowledge, some of which may not be present in models.
>
> **Experiment**: We compare PoE and MCP on 8 new tasks from BIG-Bench: 4 of them are mainly based on logical reasoning (logical_args (LA), identify_math_theorems (IMT), code_line_description (CLD), reasoning_about_coloered_objects (RACO)); the other 4 are control tasks that rely more on social or commonsense reasoning (crass_ai (CAI), evaluating_information_essentiality (EIE), riddle_sense (RS), identify_odd_metaphor (IOM)).
>
>
> **Results**: As shown below (numbers are accuracy scores), PoE outperforms MCP by large margins on all 4 logical reasoning tasks (LA, IMT, CLD, RACO), especially LA (18.8%), which is similar to improvements on LD (13.8%) and CC (12.0%). The performance on LA is convincing, because LA is made up of GRE-style logical questions, and a common strategy used by human test takers for such questions is PoE.
> On all 4 control tasks (CAI, EIE, RS, IOM), PoE underperforms MCP.
>
> | Task   |   MCP |   PoE |   PoE - MCP |
> |:-------|------:|------:|------------:|
> | LA     |  50   |  68.8 |        18.8 |
> | IMT    |  34   |  47.2 |        13.2 |
> | CLD    |  67.2 |  75.9 |         8.7 |
> | RACO   |  53.8 |  60.6 |         6.8 |
> | CAI    |  84.1 |  81.8 |        -2.3 |
> | EIE    |  25   |  19.1 |        -5.9 |
> | RS     |  55.1 |  49   |        -6.1 |
> | IOM    |  56.2 |  50   |        -6.2 |
>
>
> In addition, we report results on the same 4 logical reasoning tasks using gpt-3.5-turbo (shown below). We find PoE consistently outperforms MCP on logical reasoning tasks. This result also suggests the benefits of PoE generalize to different models.
> | Task   |   MCP |   PoE |   PoE - MCP |
> |:-------|------:|------:|----------:|
> | RACO   |   4.8 |  50.4 |      45.6 |
> | IMT    |  34   |  49.8 |      15.8 |
> | LA     |  50   |  55   |       5   |
> | CLD    |  92.4 |  91.4 |      -1   |
>
>
>
> **Conclusion**: Although PoE works on different reasoning tasks (Table 1), its performance is most significant and consistent on logical reasoning tasks.
>
> > 2. [Compared to MCP,] Why does the scoring order [of PoE] change after masking [line 536-565]?
>
> Because MCP is distracted by some wrong options. That may prevent it from selecting the right option, a problem addressed in PoE.
>
> MCP does two things (eliminate wrong answers, and choose from the remaining) in one step, and PoE does them separately in two steps. Inspired by human behavior, we assume disentangling these two steps is better than not. This assumption is supported by recent works on multi-step prompting, like [CoT](https://arxiv.org/abs/2205.11916), [ToT](https://arxiv.org/pdf/2305.10601.pdf), [GoT](https://arxiv.org/pdf/2308.09687.pdf). The change of scoring order verifies this assumption.
>
>
> > 3. … [few-shot] results [are missing] for LD or CC [to show if PoE still largely outperforms MCP].
>
> **Conclusion**: PoE seems not well-adapted to few-shot settings, but we do find PoE works well on logical reasoning tasks, compared to MCP.
>
> **Results**: As shown below, PoE is still better than MCP in the three-shot setting, but their performance gap is smaller (CC: 15.6% to 0.2%; LD: 16.2% to 1.6%). PoE does not perform well in 3-shot, and possible reasons include: 1) We do not optimize prompt or demonstrations for few-shot PoE; 2) The instructions we use in zero-shot PoE are already powerful, and few-shot demonstrations may introduce some noise.
>
> We also notice both MCP and PoE perform worse on LD in 3-shot, which we will continue to investigate.
>
>
> | Task   |   N Shot |   MCP |   PoE |   PoE - MCP |
> |:-------|---------:|------:|------:|------------:|
> | CC     |        0 |  60.4 |  76   |        15.6 |
> | CC     |        3 |  76.8 |  77   |         0.2 |
> | LD     |        0 |  39.8 |  56   |        16.2 |
> | LD     |        3 |  17.6 |  19.2 |         1.6 |
>
>
> # Questions For The Authors:
>
> > 1. Why do you mask the incorrect answers rather than simply removing them from the prompt?
>
> We do that for two reasons:
> - Our method is simpler to implement. Removing incorrect options would change the number of options, and the remaining options need to be relabeled (e.g., “ADE” to “ABC”).
> - We implement this method and compare it to our original method. We find their performance very close (<1%).
>
> > 2. Do you have any idea why the top-choice ordering changes in some cases after elimination?
>
> Yes, we do. Please check our response to the second reason to reject above.
>
> # Typos Grammar Style And Presentation Improvements:
>
> > 1. 133: What does " $T$ to 1" mean?
>
> That means “we use template $T$ for three things, and the first thing is …”
>
> We will clarify this in our next version.

---

### Official Review · Reviewer_Q4aG · 2023-08-04

**Soundness:** 3

**Excitement:**

3: Ambivalent: It has merits (e.g., it reports state-of-the-art results, the idea is nice), but there are key weaknesses (e.g., it describes incremental work), and it can significantly benefit from another round of revision. However, I won't object to accepting it if my co-reviewers champion it.

**Paper Topic And Main Contributions:**

This paper proposes an approach to improve LLM's accuracy for multiple choice questions. The approach divides the LLM reasoning process into two steps, first eliminates unlikely choices, then choose the best option among the remaining items.

**Questions For The Authors:**

1. For a question with N choices, instead of simply decompose the solution process into two steps, what if you decompose it into N-1 steps (first remove most unlikely solution, then remove the 2nd unlikely solution etc.)?
2. How does the option elimination process compare with directly picking the top k scores as candidates (i.e. si >= top_k(s1, ..., si) in Eq. 5)?
3. It remains unclear why the approach needs to replace eliminated option with the token "[MASK]". Does it matter if we use a different token to indicate invalid options?

**Reasons To Accept:**

The motivation of the paper, namely dividing the multiple-choice solution process into two steps, makes a lot of sense.

**Reasons To Reject:**

The scope of the proposed approach is very limited (only fits multiple-choice questions). What is more, it remains unclear how effective this approach is, compared with the straightforward multiple choice prompting.

**Reproducibility:**

5: Could easily reproduce the results.

**Reviewer Confidence:**

4: Quite sure. I tried to check the important points carefully. It's unlikely, though conceivable, that I missed something that should affect my ratings.

---

> ### Author Rebuttal · Authors · 2023-08-28
>
> We sincerely thank the reviewer for giving feedback and suggesting further experiments. We are encouraged that the reviewer finds our motivation sensible and our work highly reproducible (5).
>
> # Reasons to Reject:
>
> > 1. The scope of the proposed approach [PoE] is very limited (only fits multiple-choice questions).
>
> We agree that PoE is tailored for multiple-choice tasks, but we think doing so is justifiable:
> - These tasks span many NLP tasks like reading comprehension, natural language inference, text classification, and more. We think the scope is appropriate for a [short paper](https://2023.emnlp.org/calls/main_conference_papers/#short-papers:~:text=taken%20into%20account.-,Short%20Papers,-Short%20paper%20submissions), not to mention other accepted long papers dealing with similar or narrower scopes ([Surface Form Competition](https://aclanthology.org/2021.emnlp-main.564.pdf), [Calibrate Before Use](http://proceedings.mlr.press/v139/zhao21c/zhao21c.pdf)).
> - Our method could extend to generation tasks (similar to beam search, i.e., keeping likely generation and removing unlikely ones), which we would like to explore in the future.
> - We have provided evidence (in the original paper and additional experiments) for our main claim, i.e., PoE works on multiple-choice questions.
>
> >2. ... unclear how effective this approach [PoE] is, compared with the straightforward multiple choice prompting [MCP].
>
> We agree that clarity is essential. In fact, the comparison is a major focus of our paper, and we have presented it in Table 1 (last 2 columns), section 4 (line 184-199), and appendix B.2 (line 536-565). We find PoE outperforms MCP on 5 out of 8 tasks, and achieves comparable performance on the other 3. We would revise the paper to make the comparison more explicit.
>
> P.S. In response to Review dmiM (first reason to reject), we compare PoE to MCP on 8 new tasks.
>
> # Questions For The Authors:
>
> > 1. For a question with N choices … what if you decompose it into N-1 steps (first remove the most unlikely solution, then remove the 2nd unlikely solution etc.)?
>
> **Conclusion**: The proposed method (Iter PoE) achieves comparable performance to our method (PoE), but costs more time and compute. We appreciate this suggestion, but it seems PoE is a better choice.
>
> **Experiments**: We implement Iter PoE to compare with PoE on 8 tasks. The results are shown below, where numbers are accuracy scores (%).
>
> | Task   |   PoE |   Iter PoE |   PoE - Iter PoE |
> |:-------|------:|-----------:|-----------------:|
> | ANLI   |  55.6 |       55.6 |              0   |
> | CC     |  76   |       74.2 |              1.8 |
> | CQA    |  89.2 |       88.4 |              0.8 |
> | DQA    |  67.8 |       67.8 |              0   |
> | LD     |  56   |       57.2 |             -1.2 |
> | SIQA   |  82   |       82   |              0   |
> | SS     |  75.6 |       76.6 |             -1   |
> | SIT    |  23.4 |       23.8 |             -0.4 |
>
>
> > 2. How does the option elimination process compare with directly picking the top k scores as candidates (i.e. si >= top_k(s1, ..., si) in Eq. 5)?
>
> **Conclusion**: The proposed masking strategy (“Min K”) slightly underperforms the ones we used (“Below Average” and “Lowest”). “Min K” introduces a hyperparameter (k) that is task-dependent and requires tuning, whereas ours don’t. We appreciate this suggestion, but it seems our methods are better.
>
> **Experiments**:  We implement “Min K” and compare it to “Lowest”, and show the best one on each task in the following table (Min K (2) means the best value of k is 2). We find “Min K” is the best on 3 out of 8 tasks. In addition, k is task-dependent: the best value of k varies, and the effect of increasing k is uncertain (accuracy could increase or decrease).
>
> | Task | Best Mask | Accuracy |
> |:-------|:------------|-----------:|
> | ANLI | Min K (2) | 57.8 |
> | CC | Lowest | 76 |
> | CQA | Lowest | 89.5 |
> | DQA | Lowest | 67.8 |
> | LD | Lowest | 56 |
> | SIQA | Lowest | 81.7 |
> | SS | Min K (2) | 76.6 |
> | SIT | Min K (3) | 27.4 |
>
>
> > 3. Does it matter if we use a different token [other than “[MASK]”] to indicate invalid options?
>
> **Conclusion**: Yes, but there are many good replacements, and their performances are similar. Tuning this hyperparameter is time-consuming, and may not be critical to our method. Still, we appreciate this suggestion.
>
> **Experiments**: Following [Calibrate Before Use](http://proceedings.mlr.press/v139/zhao21c/zhao21c.pdf), we compare 5 mask tokens (shown below, where “empty” is an empty string, and “d…s” is a random string). Their performances are similar, and the best mask token is task-dependent.
>
> | Task | Best Mask Token | Accuracy |
> |:-------|:---------------------|-----------:|
> | ANLI | "empty" | 57.2 |
> | CC | "[N/A]" | 79.2 |
> | CQA | "dasjhasjkdhjskdhds" | 89.6 |
> | DQA | "[N/A]" | 68.2 |
> | LD | "[N/A]" | 56.8 |
> | SIQA | "[mask]" | 82.6 |
> | SS | "empty" | 78.6 |
> | SIT | "[MASK]" | 23.4 |

---

### Official Review · Reviewer_utpz · 2023-08-05

**Soundness:** 4

**Excitement:**

3: Ambivalent: It has merits (e.g., it reports state-of-the-art results, the idea is nice), but there are key weaknesses (e.g., it describes incremental work), and it can significantly benefit from another round of revision. However, I won't object to accepting it if my co-reviewers champion it.

**Paper Topic And Main Contributions:**

The paper proposes a simple trick for multi-choice QA: eliminate some choices first, then re-answer the question with eliminated choices masked. It is shown with both ChatGPT and FLAN-T5-XL on multiple datasets to work (somehow).

**Reasons To Accept:**

the technique is well motivated and reasonable, and multiple benchmarks and design choices are considered for FLAN-T5-XL.

**Reasons To Reject:**

- ChatGPT (which model, gpt-3.5-turbo?) experiment is too simple, just two datasets, and on one of them it doesn't really work well. I feel ChatGPT should be (obviously) more important than FLAN-T5-XL?

- FLAN-T5-XL across many datasets, seems some show bigger improvements, while others show very little improvement, any intuition?

- few-shot performance gain seems small overall, not sure if added computation step is worthy.

**Reproducibility:**

4: Could mostly reproduce the results, but there may be some variation because of sample variance or minor variations in their interpretation of the protocol or method.

**Reviewer Confidence:**

3: Pretty sure, but there's a chance I missed something. Although I have a good feel for this area in general, I did not carefully check the paper's details, e.g., the math, experimental design, or novelty.

---

> ### Author Rebuttal · Authors · 2023-08-28
>
> We sincerely thank the reviewer for constructive feedback. We are encouraged to find that the reviewer finds our work “well motivated and reasonable”, and appreciates our experiments and reproducibility (4).
>
> # Reasons to Reject:
>
> > 1. ChatGPT (which model, gpt-3.5-turbo?) experiment is too simple, just two datasets, and on one of them [CQA] it doesn't really work well. I feel ChatGPT should be (obviously) more important than FLAN-T5-XL?
>
> The model we used is gpt-3.5-turbo.
>
> **”ChatGPT experiment is simple”:** We agree, but we think it is justifiable:
> - We don’t intend to thoroughly compare ChatGPT and FLAN-T5 on all 8 tasks, but to highlight that our method (PoE) works on both open-sourced models like FLAN-T5 and LLMs like ChatGPT.
> - Implementing PoE for ChatGPT is non-trivial, because whereas PoE uses logits to mask options, ChatGPT does not provide logits.
> - Our budget is limited.
>
>
> **Why ChatGPT does not work well on CQA:** Possibly the prompt, as we copy the ones designed for FLAN-T5. In the following experiment, we modify the prompt, and find PoE outperforms MCP (the best baseline) on CQA (2%).
>
> **Additional Experiments:** Using gpt-3.5-turbo, we compared PoE and MCP on all 8 tasks used in our paper, as well as 4 new logical reasoning tasks (our method performs well on logical reasoning tasks, which is discussed in the response to the second reason to reject).
>
> **Results:** For the 8 original tasks (shown below, numbers are accuracy scores), PoE outperforms MCP on 5 tasks, which is consistent with the performance of FLAN-T5. This means PoE applies to LLMs like ChatGPT.
>
> | Task   |   MCP |   PoE |   PoE - MCP |
> |:-------|------:|------:|------------:|
> | SIT    |   2   |  19.4 |        17.4 |
> | ANLI   |  40.2 |  44.6 |         4.4 |
> | SS     |  79.4 |  82.8 |         3.4 |
> | CQA    |  75   |  77   |         2   |
> | LD     |  37.6 |  38   |         0.4 |
> | SIQA   |  72.6 |  71.2 |        -1.4 |
> | DQA    |  48.8 |  42.6 |        -6.2 |
> | CC     |  79.6 |  66.8 |       -12.8 |
>
> Furthermore, PoE outperforms MCP by large margins on 4 logical reasoning tasks (shown below), which further corroborates our assumption that PoE performs well on logical reasoning tasks.
>
> | Task   |   MCP |   PoE |   PoE - MCP |
> |:-------|------:|------:|------------:|
> | RACO   |   4.8 |  50.4 |        45.6 |
> | IMT    |  34   |  49.8 |        15.8 |
> | LA     |  50   |  55   |         5   |
> | CLD    |  92.4 |  91.4 |        -1   |
>
>
> We will continue to work on PoE for LLMs in the next version.
>
> **”ChatGPT is more important than FLAN-T5-XL”:** We agree that ChatGPT is one of the most important and powerful LLMs. Nevertheless, it is close-sourced and updated frequently, which is not ideal for interpretability and reproducibility. Also, our main focus is open-sourced LMs, which are prevalent and widely used by both academia and industry. We think we have provided sufficient evidence to support our claim, i.e., PoE works on open-sourced models like FLAN-T5.
>
> > 2. FLAN-T5-XL across many datasets, seems some show bigger improvements, while others show very little improvement, any intuition?
>
> **Intuition:** PoE works well on tasks that mainly require logical reasoning, which is  verified on 8 new tasks (This is a common concern shared by reviewer dmiM. Please refer to that response (first reason to reject) for more details).
>
> **Results:** As shown below, PoE largely outperforms MCP on the four logical reasoning tasks (first four rows); PoE underperforms MCP on the other 4 tasks, all of which require social or commonsense reasoning. One explanation is that logical reasoning is a general skill, whereas social or commonsense reasoning relies on specific knowledge, some of which may not be present in language models.
>
>
> | Task   |   MCP |   PoE |   PoE - MCP |
> |:-------|------:|------:|------------:|
> | LA     |  50   |  68.8 |        18.8 |
> | IMT    |  34   |  47.2 |        13.2 |
> | CLD    |  67.2 |  75.9 |         8.7 |
> | RACO   |  53.8 |  60.6 |         6.8 |
> | CAI    |  84.1 |  81.8 |        -2.3 |
> | EIE    |  25   |  19.1 |        -5.9 |
> | RS     |  55.1 |  49   |        -6.1 |
> | IOM    |  56.2 |  50   |        -6.2 |
>
>
> > 3. Few-shot performance gain seems small overall, not sure if added computation step is worthy.
>
> We agree that few-shot performance gain is small. Still, we think the few-shot experiment is worthy and necessary, because it shows our method generalizes beyond zero-shot. We will continue to improve few-shot performance in the next version.

---

### Meta-Review · Area_Chair_YeLa · 2023-09-19

**Recommendation:** 4

**Metareview:**

The authors proposed a two-step scoring method for multiple choice reasoning tasks, based on the process of elimination.
The 2 out of 3 reviewers are selected strong soundness, and reproducibility is enough.

---

### Decision · Program_Chairs · 2023-10-07

**Decision:**

Accept-Main

**Comment:**

The authors proposed a two-step scoring method for multiple choice reasoning tasks, based on the process of elimination.
The 2 out of 3 reviewers are selected strong soundness, and reproducibility is enough.